

# Development of multiplex real-time PCR for simultaneous detection of common fungal pathogens in invasive mycoses

Yasmin Khairani Muhammad Ismadi[1], Suharni Mohamad[2] and
Azian Harun[1,3]

[1] Department of Medical Microbiology and Parasitology, School of Medical Sciences, Universiti
Sains Malaysia, Kubang Kerian, Kelantan, Malaysia
[2] School of Dental Sciences, Universiti Sains Malaysia, Kubang Kerian, Kelantan, Malaysia
[3] Hospital Universiti Sains Malaysia, Universiti Sains Malaysia, Kubang Kerian, Kelantan, Malaysia

## ABSTRACT

**Background:** Fungi are common opportunistic pathogens that pose a significant
threat to immunocompromised patients, particularly when late detection occurs.
**Methods:** In this study a multiplex real-time PCR has been developed for
simultaneous detection of common fungal pathogens associated with invasive
mycoses in a diagnostic setting.
**Results:** The specificity of the assay was rigorously tested on 40 types of organisms
($n = 65$), demonstrating 100% specificity. The limit of detection was determined to be
100 pg/μl ($10^6$ copies/μl), achievable within a rapid 3-h timeframe. The PCR assay
efficiency exhibited a range between 89.77% and 104.30% for each target organism,
with linearity falling between 0.9780 and 0.9983.
**Conclusion:** This multiplex real-time PCR assay holds promise for enhancing the
timely and accurate diagnosis of invasive mycoses, particularly in
immunocompromised patient populations.

## INTRODUCTION

As the prevalence of immunosuppressive conditions continues to rise, infections caused by
*Aspergillus fumigatus*, *Aspergillus terreus*, *Aspergillus niger*, *Candida albicans*, *Candida
glabrata*, and *Cryptococcus neoformans* present substantial threats to vulnerable
individuals, predominantly afflicting immunocompromised patients, and pose a
significant concern due to their potentially fatal outcomes (*Casadevall et al., 2019*). This
vulnerable population is at an increased risk of developing these infections, and delays in
diagnosis can significantly impact prognosis (*Pathakumari, Liang & Liu, 2020*). The
urgency for rapid and precise identification of fungal infections has intensified with the
escalating morbidity and mortality associated with invasive mycoses each year
(*Denning, 2024*; *Pathakumari, Liang & Liu, 2020*; *Rayens, Norris & Cordero, 2022*) and
*Aspergillus* and *Candida* species continue to be the primary culprits in invasive mycoses
(*Bajpai et al., 2019*).

Corresponding author
Azian Harun, azian@usm.my

Timely diagnosis is essential for effective intervention, guiding treatment strategies, and improving patient outcomes. Conventional microscopy and molecular methods, such as polymerase chain reaction (PCR), are the current tools used to identify fungal pathogens (*Kidd et al., 2019*). PCR is particularly valuable in molecular identification due to its highly specific detection of fungal DNA. However, the identification of filamentous fungi, which are known for causing invasive mycoses, remains a time-consuming process, especially when relying on conventional sequencing methods (*Fang et al., 2023*).

Nucleic acid amplification tests (NAATs), especially multiplex real-time PCR, offer significant advantages in detecting fungal pathogens associated with invasive mycoses. These molecular techniques enable rapid, sensitive, and specific identification of multiple fungal species simultaneously, addressing the limitations of traditional culture-based methods (*Arvanitis et al., 2014*). NAATs can detect fungal pathogens earlier in the infection process, even when viable organisms are no longer present, potentially improving patient outcomes through earlier diagnosis and treatment (*Fuchs, Lass-Florl & Posch, 2019*). The high-throughput capability of multiplex real-time PCR allows for efficient screening of multiple targets in a single reaction, reducing both time and resource requirements (*Valero et al., 2016*). Moreover, NAATs' ability to accurately detect and differentiate closely related fungal species is particularly valuable in managing invasive mycoses, where rapid and precise identification is crucial for appropriate antifungal therapy (*Donnelly et al., 2019*). Additionally, the quantitative aspect of real-time PCR can offer insights into fungal burden, aiding in monitoring treatment response and disease progression (*White et al., 2015*).

Therefore, to overcome the identified shortcomings, this study aimed to develop a rapid assay for simultaneous detection method for *A. fumigatus*, *A. terreus*, *C. albicans* and *C. glabrata*. The approach includes the incorporation of an internal amplification control and the performance of analytical validation for sensitivity and specificity on the developed assay.

## MATERIALS AND METHODS

### Fungal and bacterial strains and cultures

The fungal isolates used in this study were representative of clinical and reference strains obtained from the Mycology Laboratory, Department of Medical Microbiology and Parasitology, Hospital Universiti Sains Malaysia, and the American Type Culture Collection (ATCC, Manassas, Virginia, USA). The representative strains underwent ribosomal RNA gene sequencing for species confirmation prior to the evaluation. A total of 65 organisms were used in this study, which comprised 55 fungal strains (11 *Aspergillus* species, 27 *Candida* species, 17 other fungal species), and 10 bacterial strains (Table S1). Fungal isolates were cultured on Sabouraud dextrose agar (SDA) medium (Oxoid, Basingstoke, UK) at 30 °C for 2 to 7 days while the bacterial isolates were cultured on Mueller Hinton Agar (Oxoid, Basingstoke, UK) at 37 °C overnight.

## DNA extraction

The fungal isolates were cultivated on SDA plates at a temperature of 30 °C for a duration of 2 to 7 days. Conidia and hyphae were collected and harvested at −80 °C until the DNA extraction process. Both types of strains were extracted using the phenol-chloroform method. Briefly, a mixture of 500 μl lysis buffer (composed of 1.0% SDS, NaCl, 0.5 M EDTA, 1 M Tris-HCl (pH 8)) along with 5 μl 2-mercaptoethanol (Sigma-Aldrich, Burlington, MA, USA) was added into frozen fungal mycelia and yeast colonies, which have been crushed using a pellet pestle. After vigorous vortexing, the mixture was incubated at 65 °C for 90 min. After incubation, a solution of phenol:chloroform:isoamyl alcohol (25:24:1) (Sigma Aldrich, Burlington, MA, USA) was added to the mixture and mixed by inverting the tubes for 2 min. The mixture was then subjected to centrifugation at a speed of $20,000 \times g$ for 5 min. The uppermost aqueous layer was transferred to a new 1.5 ml tube and mixed with an equal volume of isopropanol, allowing precipitation for a minimum of 2 h. The resultant mixture was then centrifuged again at $20,000 \times g$ for 5 min, resulting to the formation of DNA pellets at the bottom of the tube. Next, a washing step was repeated twice with 70% (v/v) ethanol. Lastly, the DNA pellets were air-dried at room temperature followed by resuspension in sterile nuclease-free water. The extracted DNA was subsequently stored at a temperature of −20 °C. Meanwhile, bacterial isolates were extracted using the DNeasy Blood & Tissue kit (Qiagen Inc., Hilden, Germany) according to the manufacturer's protocols. The nucleic acid quantification was performed using a microplate spectrophotometer (Thermo Fisher Scientific, Waltham, MA, USA).

## Oligonucleotide pairs design

Species-specific oligonucleotide pairs (primers and probes) were designed using the IDT DNA PrimerQuest online tool (https://sg.idtdna.com/PrimerQuest/), relying on the housekeeping genes with GenBank accession numbers: *bgt1* gene (AF038596), *benA* gene (KF669507), ITS2 gene (AJ853768), *LEU2* gene (CP048125). Primer specificity was assessed *in silico* using Primer-BLAST for the selected DNA regions. Evaluation included measuring homodimer, heterodimer, and hairpin formation of primers for simultaneous detection, ensuring heterodimer delta G values were ≤−9 kcal/mol for all primers. However, five primer pairs exhibited ΔG values ranging from −10.21 to −11.27 kcal/mole (Table S2A). Additionally, seven probe-primer pairs with ΔG values between −9.19 to −12.76 kcal/mole (Table S2B) and two probe-probe pairs demonstrating a ΔG value of −11.42 kcal/mole (Table S2C). These primers and probes were subsequently synthesized by IDT, Singapore, and their corresponding amplicon sizes are detailed in Table 1.

## Quantitative analysis of multiplex real-time PCR

Quantitative analysis was performed using the BioRad CFX96 Touch™ Real-Time PCR Detection System (BioRad, Hercules, CA, USA) using SsoAdvanced Universal Probes Supermix (BioRad, Hercules, CA, USA) and quantification cycle numbers were analyzed *via* BioRad CFX Manager software. The primer and probe concentrations used in this assay were 500 and 200 nM (Tables S3A–S3D and Tables S4A–S4D), respectively. The amplification protocol was conducted as follows: 3 min of initial denaturation at 95 °C

**Table 1 List of oligonucleotide sequences designed for multiplex identification.**

| Targeted organisms | Target loci | Oligonucleotide sequences (5′-3′) | GC content (%) | Tm (°C) | References |
|---|---|---|---|---|---|
| *A. fumigatus* | β-1,3 glucanosyl-transferase (*bgt1*) | bgt1_F- GCT GCT GCC TCC AAG AAT G | 57.9 | 56.8 | *Ismadi, Mohamad & Harun (2023)* |
| | | bgt1_R- GCA GTC ACT CGC GGA GTA G | 63.2 | 57.8 | |
| | | **FAM**-CCT GGG CAA TAA GAA CGA AGG CG-**BHQ1** | 56.5 | 60.5 | This study |
| *A. terreus* | β-tubulin (*benA*) | benA_F- GGC TCC CAT AAT GGA GGT TTA C | 50.0 | 55.3 | *Ismadi, Mohamad & Harun (2023)* |
| | | benA_R- GTG AAG AAT CTG TCC CAG GAT G | 50.0 | 55.2 | |
| | | **TEXRD**-AAA CAG CTT CAA TGG CTC CTC CGA-**BHQ2** | 50.0 | 60.8 | This study |
| *C. albicans* | Internal transcribed spacer 2 (ITS2) | ITS2_F- GTT TGC TTG AAA GAC GGT AGT G | 45.5 | 54.3 | *Ismadi, Mohamad & Harun (2023)* |
| | | ITS2_R- AAG ATA TAC GTG TGG ACG TTA C | 40.9 | 52.0 | |
| | | **HEX**-TAA GGC GGG ATC GCT TTG ACA ATG G-**BHQ1** | 52.0 | 61.4 | This study |
| *C. glabrata* | Isopropylmalate dehydrogenase (*LEU2*) | LEU2_F-GTT AGA GAA CTA GTG GGT GGT ATT T | 40.0 | 54.1 | *Ismadi, Mohamad & Harun (2023)* |
| | | LEU2_R-TAG GTA AAG GTG GGT TGT GTT G | 45.5 | 54.8 | |
| | | **CY5.5**-AAG ATG AAG GTG ATG GTG TCG CCT-**BHQ2** | 50.0 | 60.4 | This study |
| *H. pylori* (*IAC) | Phosphoglucosamine mutase (*glmM*) | glmM_F-AAC TTA TCC CCA ATC GCG CA | 50.0 | 57.1 | This study |
| | | glmM_R-GCC CTT TCT TCT CAA GCG GT | 55.0 | 57.9 | This study |
| | | **CY5**-AGG GCT AAA TTG CTC ATG TTA GTA GC-**BHQ2** | 42.3 | 57.1 | This study |

**Note:**
* IAC–internal amplification control.

followed by 40 cycles of denaturation at 95 °C for 15 s, annealing and elongation at 60 °C for 30 s.

The quantification cycle (Cq) was calculated by the software. The baseline threshold was set at 50 and 100 RFU for all channels/fluorophores, and any Cq ≤ 40 is considered positive. A standard curve was obtained using the measured results, and the slope value, mean Cq value (MCq), and standard deviation (Std) were calculated. The PCR efficiency (E value, %) was calculated by the following equation: $[10^{(-1/(\text{slope}))} - 1] \times 100$.

## Analytical sensitivity and specificity

To evaluate LOD, a series of 10-fold dilutions of DNA templates from the reference strains were prepared, ranging from 100 ng/µl to 1 pg/µl. Real-time PCR reactions were set up using two microlitres of each dilution in triplicate. Standard curves were constructed based on the mean of Cq values and $\log_{10}$ (DNA copies). DNA copy numbers were calculated using online tool provided by the URI Genomics & Sequencing Centre (https://web.uri. edu/riinbre/mic/). These limits are established by testing samples with known

concentrations and identifying the lowest concentration at which the assay reliably performs. To calculate the conversion factor from copies per reaction to copies per millilitre, it is essential to know the reaction volume used in the real-time PCR assay. The calculation assumes that all the DNA copies present in the reaction volume are representative of the total volume (typically 0.02 ml for many PCR reactions): DNA copies/ml = DNA copies per reaction/reaction volume (ml).

Analytical specificity test was carried out on American Type Culture (ATCC) and clinical strains which a total of 12 ATCC strains and 53 clinical strains, including non-target isolates, were included in the study. Two microlitres of DNA extracted from these isolates were used as templates for evaluating analytical sensitivity and specificity in the developed multiplex real-time PCR assay.

### Assay precisions

In the process of developing a multiplex real-time PCR assay, intra-assay evaluation refers to the assessment of the performance of the assay within a single experimental run. This evaluation involves analysing the consistency and reliability of results obtained within the same run and this assay was prepared in triplicates. All protocols were followed the guidelines outlined in the Minimum Information for the Publication of Real-Time Quantitative PCR Experiments (MIQE) (*Bustin et al., 2009*).

## RESULTS

### Qualitative analysis of analytical specificity

The analytical specificity of the developed multiplex real-time PCR was determined using 65 strains, as presented in Table 2. No amplification occurred in non-targeted isolates. All positive control DNA was successfully amplified in all oligonucleotides designed for this assay and there was no interference between probes. Specifically, three *A. fumigatus* isolates showed amplification of *bgt1* gene, three *A. terreus* isolates showed amplification of *benA* gene, 14 *C. albicans* isolates showed amplification of ITS2 gene and eight *C. glabrata* isolates showed amplification of *LEU2* gene. The Cq values for targeted organisms ranged from 20.14 to 24.67 for *A. fumigatus*, 17.63 to 28.90 for *A. terreus*, 16.88 to 32.90 for *C. albicans*, and 20.07 to 21.11 for *C. glabrata*, respectively. Additionally, the internal amplification control (IAC) was successfully detected in other organisms, indicating the overall success of the assay and the absence of PCR inhibitors. Figure 1 visually represents the non-targeted organisms in the analytical test of the developed multiplex real-time PCR assay.

### Quantitative analysis of analytical sensitivity

In assessing the analytical sensitivity of the developed multiplex real-time PCR assay, the limit of detection (LOD) was determined for all targeted genes (*bgt1*, *benA*, ITS2 and *LEU2*). No interference occurred during the amplification of four oligonucleotides analyzed in the multiplex assay, and the DNA was successfully amplified from concentrations ranging from 100 ng/µl to 100 pg/µl. The LOD was determined based on the lowest fungal genomic DNA concentration (100 pg/µl). For all targeted organisms (*A. fumigatus*, *A. terreus*, *C. albicans*,

**Table 2 List of organisms tested for analytical specificity of the developed multiplex real-time PCR assay.**

| No | | Organism | Reference strains, $n$ | Clinical strains, $n$ | Total test, $n$ | Positive test, $n$ (%) | Negative test, $n$ (%) |
|---|---|---|---|---|---|---|---|
| 1. | Targeted organism | A. fumigatus | 1 | 2 | 3 | 3 (100) | 0 (0) |
| 2. | | A. terreus | 0 | 3 | 3 | 3 (100) | 0 (0) |
| 3. | | C. albicans | 1 | 13 | 14 | 14 (100) | 0 (0) |
| 4. | | C. glabrata | 0 | 8 | 8 | 8 (100) | 0 (0) |
| 5. | Yeast | C. tropicalis | 0 | 1 | 1 | 0 (0) | 1 (100) |
| 6. | | C. parapsilosis | 0 | 1 | 1 | 0 (0) | 1 (100) |
| 7. | | C. krusei | 1 | 0 | 1 | 0 (0) | 1 (100) |
| 8. | | C. lusitaniae | 0 | 1 | 1 | 0 (0) | 1 (100) |
| 9. | | C. dubliniensis | 0 | 1 | 1 | 0 (0) | 1 (100) |
| 10. | | C. neoformans | 0 | 1 | 1 | 0 (0) | 1 (100) |
| 11. | | R. mucilaginosa | 0 | 1 | 1 | 0 (0) | 1 (100) |
| 12. | | T. mycotoxinovorans | 0 | 1 | 1 | 0 (0) | 1 (100) |
| 13. | Mold | A. flavus | 0 | 1 | 1 | 0 (0) | 1 (100) |
| 14. | | A. niger | 0 | 1 | 1 | 0 (0) | 1 (100) |
| 15. | | A. nidulans | 0 | 1 | 1 | 0 (0) | 1 (100) |
| 16. | | A. lentulus | 0 | 2 | 2 | 0 (0) | 2 (100) |
| 17. | | E. dermatitidis | 0 | 1 | 1 | 0 (0) | 1 (100) |
| 18. | | F. solani | 0 | 1 | 1 | 0 (0) | 1 (100) |
| 19. | | G. candidum | 0 | 1 | 1 | 0 (0) | 1 (100) |
| 20. | | L. prolificans | 0 | 1 | 1 | 0 (0) | 1 (100) |
| 21. | | M. canis | 0 | 1 | 1 | 0 (0) | 1 (100) |
| 22. | | M. guillermondii | 0 | 1 | 1 | 0 (0) | 1 (100) |
| 23. | | M. gypseum | 0 | 1 | 1 | 0 (0) | 1 (100) |
| 24. | | Mucor spp. | 0 | 1 | 1 | 0 (0) | 1 (100) |
| 25. | | P. aurantiacum | 0 | 1 | 1 | 0 (0) | 1 (100) |
| 26. | | P. variotii | 0 | 1 | 1 | 0 (0) | 1 (100) |
| 27. | | R. microsporus | 0 | 1 | 1 | 0 (0) | 1 (100) |
| 28. | | S. schenckii | 0 | 1 | 1 | 0 (0) | 1 (100) |
| 29. | | T. marneffei | 0 | 1 | 1 | 0 (0) | 1 (100) |
| 30. | | T. rubrum | 0 | 1 | 1 | 0 (0) | 1 (100) |
| 31. | Bacteria | B. pseudomallei | 0 | 1 | 1 | 0 (0) | 1 (100) |
| 32. | | E. coli | 1 | 0 | 1 | 0 (0) | 1 (100) |
| 33. | | E. faecalis | 1 | 0 | 1 | 0 (0) | 1 (100) |
| 34. | | K. pneumoniae | 1 | 0 | 1 | 0 (0) | 1 (100) |
| 35. | | N. meningitis | 1 | 0 | 1 | 0 (0) | 1 (100) |
| 36. | | S. aureus | 1 | 0 | 1 | 0 (0) | 1 (100) |
| 37. | | S. epidermidis | 1 | 0 | 1 | 0 (0) | 1 (100) |
| 38. | | S. typhii | 1 | 0 | 1 | 0 (0) | 1 (100) |
| 39. | | P. aeruginosa | 1 | 0 | 1 | 0 (0) | 1 (100) |
| 40. | | V. parahaemolyticus | 1 | 0 | 1 | 0 (0) | 1 (100) |

**Note:**
Specificity evaluation of the developed multiplex real-time PCR assay with a total test, $n$: 65, total positive: 28/28 (100%), total negative: 37/37 (100%) and test accuracy: 100%.

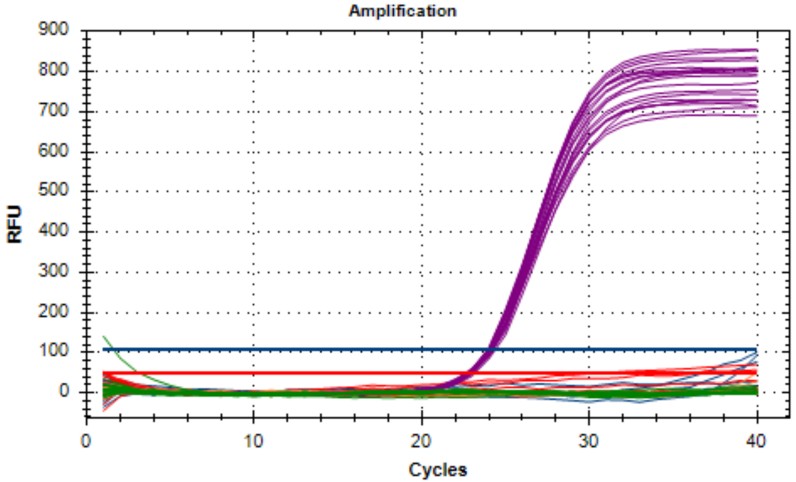

**Figure 1 The non-targeted organisms of the analytical specificity test on the developed multiplex assay.** Only the IAC (Cy5 fluorescent signal) amplified which indicates that the assay was stable.

**Table 3 Mean threshold value (Cq), standard deviation and coefficient of variation (CV) of multiplex real-time PCR assay for each targeted organism.**

| Organism | Concentration | DNA copies per reaction | DNA copies/ml | Mean Cq | *Std | *CV% |
|---|---|---|---|---|---|---|
| *A. fumigatus* | 100 pg | $3.18 \times 10^6$ | $1.59 \times 10^8$ | 35.09 | 0.30 | 0.86 |
| *A. terreus* | 100 pg | $3.05 \times 10^6$ | $1.53 \times 10^8$ | 38.54 | 1.01 | 2.64 |
| *C. albicans* | 100 pg | $6.62 \times 10^6$ | $3.31 \times 10^8$ | 34.89 | 0.85 | 2.44 |
| *C. glabrata* | 100 pg | $7.41 \times 10^6$ | $3.71 \times 10^8$ | 33.49 | 0.41 | 1.23 |

**Note:**
* Std, standard deviation; CV, coefficient of variation.

and *C. glabrata*) and their respective genes (*bgt1*, *benA*, ITS2 and *LEU2*), the LOD of the multiplex real-time PCR for detection was consistently determined to be 100 pg/µl (Table 3). This concentration is equivalent to $1.59 \times 10^6$, $1.53 \times 10^6$, $3.31 \times 10^6$ and $3.71 \times 10^6$ copies/µl, respectively. The LOD for all targeted genes remained consistent at 100 pg/µl for fungal genomic DNA.

## Efficiency, linearity of multiplex real-time PCR assay

The standard curve was generated by plotting log DNA copies/ml on the X-axis against mean Cq values on the Y-axis (Figs. 2A–2D), resulted in slope values and linearity ($R^2$) within the ranges of −3.223 to −3.594 and 0.978 to 0.9983 (Table 4). The percentages of PCR efficiency, E value, were determined to be between 89.77% and 104.3% for the multiplex assay.

## Intra-assay evaluation

Intra-assay repeatability was assessed in triplicate for the developed multiplex real-time PCR. Two concentration scales, 100 ng (highest detection limit) and 0.1 ng (lowest detection limit), were used in this evaluation, as detailed in Table 5. The intra-assay

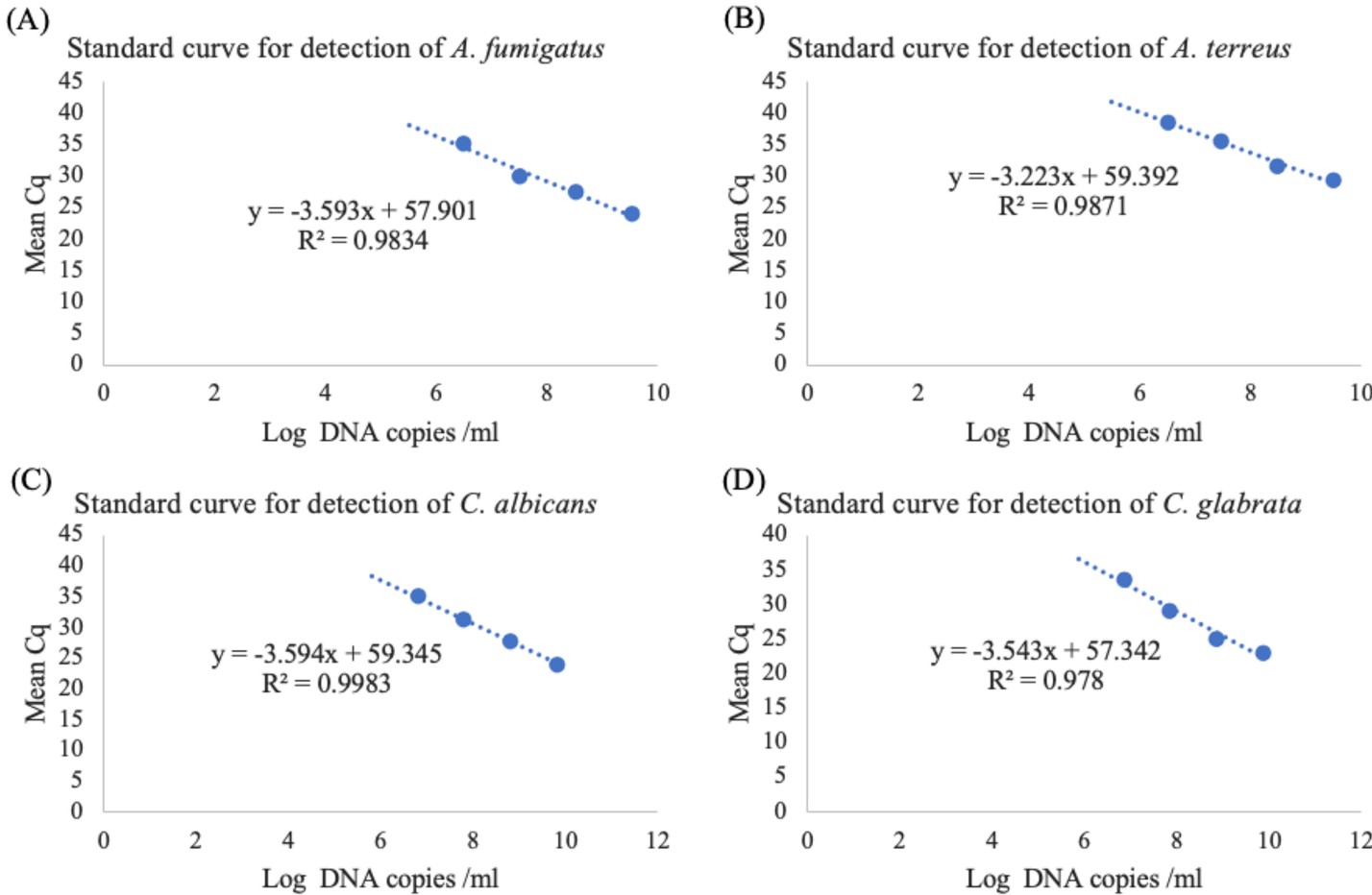

**Figure 2 Standard curve of the developed real-time PCR for the detection of (A)** *A. fumigatus*, **(B)** *A. terreus*, **(C)** *C. albicans* **and (D)** *C. glabrata*.

**Table 4 Performance of multiplex real-time PCR assays for detection of** *A. fumigatus, A. terreus, C. albicans* **and** *C. glabrata*.

| Parameter | Targeted organisms | | | |
|---|---|---|---|---|
| | *A. fumigatus* | *A. terreus* | *C. albicans* | *C. glabrata* |
| Slope | −3.593 | −3.223 | −3.594 | −3.543 |
| E value* (%) | 89.81 | 104.30 | 89.77 | 91.53 |
| Linearity | 0.9834 | 0.9871 | 0.9983 | 0.978 |

**Note:**
 * E value–PCR efficiency

coefficient of variation (CV) demonstrated an overall range of 0.29% to 2.64%. Specifically, for *A. fumigatus*, the CVs were 0.29% and 0.86% at 100 ng and 0.1 ng, respectively. For *A. terreus*, the CVs were 0.97% and 2.64%. Regarding *C. albicans*, the CVs were 0.54% and 2.44% and for *C. glabrata*, the CVs were 0.31% and 1.23%. The analytical performance was enhanced when utilizing the highest concentration of 100 ng of DNA, and this

**Table 5 Intra-assay evaluation of the multiplex real-time PCR.**

| Organism | DNA concentration (ng) | Intraassay | | |
|---|---|---|---|---|
| | | Mean Cq | Std* | CV* (%) |
| *A. fumigatus* | 100 | 24.02 | 0.07 | 0.29 |
| | 0.1 | 35.09 | 0.30 | 0.86 |
| *A. terreus* | 100 | 29.19 | 0.28 | 0.97 |
| | 0.1 | 38.54 | 1.02 | 2.64 |
| *C. albicans* | 100 | 23.97 | 0.13 | 0.54 |
| | 0.1 | 34.90 | 0.85 | 2.44 |
| *C. glabrata* | 100 | 23.02 | 0.07 | 0.31 |
| | 0.1 | 33.49 | 0.41 | 1.23 |

**Note:**
  * Std, standard deviation; CV, coefficient of variation.

performance of assay was not compromised even when testing the lowest detection limit of DNA concentration.

## DISCUSSION

In this study, we developed a molecular detection method for common filamentous ascomycetes *A. fumigatus* and *A. terreus*, as well as important yeast strains *C. albicans* and *C. glabrata*. The selection of these organisms for this assay was driven by their clinical significance and unique characteristics. *A. fumigatus* and *C. albicans* were chosen as they are the most common fungal pathogens in invasive mycoses, accounting for a significant proportion of fungal infections in immunocompromised patients (*Pappas et al., 2018*). The inclusion of *A. terreus* was motivated by its intrinsic resistance to amphotericin B, a characteristic that complicates treatment and necessitates accurate identification for appropriate antifungal therapy (*Rybak, Fortwendel & Rogers, 2019*). Similarly, *C. glabrata* was selected due to its increasing prevalence and reduced susceptibility to azoles and echinocandins, which poses significant challenges in clinical management (*Perlin, Rautemaa-Richardson & Alastruey-Izquierdo, 2017*; *Pfaller et al., 2019*). The rising incidence of *C. glabrata* infections, particularly in healthcare settings, further underscores the importance of its inclusion in the assay. By targeting these four species, the assay aims to cover a broad spectrum of clinically relevant fungal pathogens, including those with antifungal resistance profiles that demand careful consideration in treatment strategies (*Wiederhold, 2017*).

By targeting specific genes, this method reduces time to diagnosis compared to conventional culture methods, which can be time-consuming (*Kim et al., 2020*). The aim is to establish a robust multiplex real-time PCR assay with high specificity and sensitivity. The selected primer and probe pairs were designed at the species level, enabling the analysis and detection of specific invasive mycoses, which might not be possible with ITS sequencing. In addition, newly designed fluorescent hydrolysis probes were assessed in this study. The inclusivity of the primer and probe pairs were confirmed through both *in silico* BLAST analysis and *in vitro* multiplex real-time PCR assay enabling the simultaneous

detection of all four targets. These primer-probe pairs designed for the assay were then subjected to optimization, focusing on characteristics such as product length, secondary structure of the sequences (hairpin, homo- and heterodimer formation), GC content, and melting temperature.

Besides, an internal amplification control (IAC) was incorporated into this assay to monitor its performance and detect potential issues such as PCR inhibition or other factors that could compromise the accuracy of the results (*Moldovan & Moldovan, 2020*). The inclusion of an IAC is crucial for distinguishing between true negative results, where no target is present in the sample, and false negatives caused by technical problems (*Burbank & Ortega, 2018*). This addition ensures the validity of the results and enhances overall quality control in the developed assay. The *glmM* gene from *Helicobacter pylori* was strategically selected as the IAC due to its consistency across strains, ensuring specificity in molecular detection. Additionally, the compatibility of the *glmM* gene with the developed assay provides quantitative data, aligning with the needs of diagnostic applications. Incorporating the *glmM* gene as an IAC serves several critical functions: it enables effective monitoring of the assay, ensures proper functionality, and helps identify potential inhibitors. This approach is vital for detecting false negatives caused by sample inhibitors, as the failure of the exogenous IAC to amplify indicates underlying issues in the assay process (*Mittelberger et al., 2020*).

For the molecular identification of fungal especially filamentous ascomycetes, there is a requirement to enhance the processes and purity of DNA extraction (*Kim et al., 2020*). The quality of extracted DNA and the presence of PCR inhibitors may pose challenges in conducting quantitative assays and assessing the sensitivity of the assay. In this study, we employed a conventional phenol-chloroform extraction method without additional mechanical lysis, such as bead-beating, making the process more practical and time-efficient for diagnostic settings. This contrasts with the approach used by *Kim et al. (2020)* who faced challenges in achieving sufficient DNA purity for quantitative analysis due to their use of freeze-thawing and bead-beating methods alongside enzyme and Qiagen purification kits. Another study demonstrated that a chemical lysis method, involving detergents and proteinase K, could effectively extract DNA from both yeast and filamentous fungi, with yields comparable to those achieved with bead-beating (*Fredricks, Smith & Meier, 2005*).

In this multiplex real-time PCR assay, integrating the IAC exhibited successful amplifications with acceptable slope values ($-3.223$ to $-3.594$ and $0.978$ to $0.9983$). In standard curve analysis, a slope within range of $-3.6$ to $-3.1$ ensures consistent and reproducible results, necessitating optimization of reaction conditions for accurate and reliable PCR quantification (*Jiang et al., 2022*). The linearity of the multiplex assay was evaluated by using serial tenfold dilutions of each target sample, ranging from 100 ng/μl to 10 pg/μl. Confirmatory data indicated no cross-interference among the primers and probes used. A mixture of DNA templates from all four target fungi produced standard curves with $R^2$ and efficiency (E) values within the acceptable range of 90% to 110% (*Maza-Marquez et al., 2018*). To ensure optimal performance, we precisely adjusted the primer and probe concentrations, thereby reducing cross-reactivity and enhancing

amplification efficiency. This was further validated by the consistent linearity and E values observed in the standard curves.

Though two E values were slightly lower than 90%, at 89.81% and 89.77%, but this did not hinder the efficiency of the assay in detecting the samples. The LOD for *A. fumigatus*, *A. terreus*, *C. albicans*, and *C. glabrata* remained consistent at 100 pg/μl ($10^6$ copies/μl). This result differs slightly from previous studies, which detected *Aspergillus* species with as low as 30 fg and 40 ag/μl of DNA in their developed assays (*Johnson et al., 2012*; *Kim et al., 2020*). The reliability of the LOD depends on the specific requirements; in some cases, an LOD in the range of thousand copies/ml is considered acceptable, particularly for assays targeting genomic DNA. Achieving an acceptable LOD is crucial in fungal multiplex real-time PCR assays as it directly impacts the sensitivity and diagnostic accuracy of the assay. An LOD that is too high could result in false-negative results, especially in cases where the fungal load is low, such as in early-stage infections or in samples from immunocompromised patients (*White et al., 2015*). Conversely, a low LOD enhances the assay's ability to detect minimal quantities of target DNA, which is critical for early diagnosis and timely treatment of invasive fungal infections (IFIs). Recent studies underscore the importance of optimizing LOD in fungal assays to ensure their clinical utility. For example, White et al. emphasized that an LOD of 100 to 1,000 copies/ml is often necessary to reliably detect fungal pathogens in blood samples, where fungal DNA load is typically low (*White et al., 2015*).

In this study, analytical specificity was evaluated using diverse set of 65 organisms which consist of both fungi and bacteria and has demonstrated exceptional accuracy. The multiplex real-time PCR assay correctly identified all 28 targeted strains–*A. fumigatus*, *A. terreus*, *C. albicans*, and *C. glabrata*, with no amplification in the other 37 non-targeted strains. This underscores the specificity of the assay and ability to differentiate between closely related fungal species.

In the intra-assay evaluation, two concentration scales (100 ng/ml and 0.1 ng/ml) revealed an overall coefficient of variation (CV) range 0.29% to 2.64%. Individual fungal species showed varying CVs, but the analytical performance remained robust, emphasizing the reliability of the assay across a wide range of DNA concentrations. The observed CV range indicates minimal variability within triplicate measurements, affirming the precision of this assay.

The original aim of this present study was to demonstrate the establishment, optimization, and validation of the novel multiplex real-time RT-PCR assay. While the evaluation and validation across clinical samples have not yet been conducted, the developed assay has undergone thorough optimization, standardization, and analytical validation, including assessments of sensitivity, specificity, efficiency, and linearity. Additionally, intra-assay evaluations were performed to ensure the reliability of the assay.

## CONCLUSIONS

In conclusion, our study successfully developed and validated a robust multiplex real-time PCR assay for the molecular detection of clinically important fungi *A. fumigatus*, *A. terreus*, *C. albicans*, and *C. glabrata*. The assay, designed at the species level, offers a

significant reduction in diagnosis time compared to traditional culture methods. Rigorous validation included *in silico* analysis, *in vitro* assays, and the incorporation of an IAC for quality control. The selected *glmM* gene from *H. pylori* enhances molecular detection specificity. Despite using a practical phenol-chloroform DNA extraction method, the assay demonstrated high sensitivity, specificity, and reliability across various DNA concentrations. Analytical specificity was confirmed against a diverse set of organisms, and intra-assay evaluation indicated minimal variability. The integration of an IAC ensures result validity and enhances quality control. Overall, our multiplex real-time PCR assay proves promising application for clinical diagnostics, combining efficiency with accuracy and reliability in routine diagnostic applications.

## ACKNOWLEDGEMENTS

We would like to thank Rosmaniza Abdullah for her contribution in procuring the isolates.

### Funding
This study has been funded by Universiti Sains Malaysia RUI Grant 1001/PPSP/812206 awarded to Azian Harun. The funders had no role in study design, data collection and analysis, decision to publish, or preparation of the manuscript.

### Grant Disclosures
The following grant information was disclosed by the authors:
Universiti Sains Malaysia RUI: 1001/PPSP/812206.

### Competing Interests
The authors declare that they have no competing interests.

### Author Contributions
- Yasmin Khairani Muhammad Ismadi conceived and designed the experiments, performed the experiments, analyzed the data, prepared figures and/or tables, authored or reviewed drafts of the article, and approved the final draft.
- Suharni Mohamad conceived and designed the experiments, authored or reviewed drafts of the article, and approved the final draft.
- Azian Harun conceived and designed the experiments, authored or reviewed drafts of the article, and approved the final draft.

### Data Availability
The list of oligonucleotide sequences designed for multiplex identification is available in Table 1.

### Supplemental Information
Supplemental information for this article can be found online at http://dx.doi.org/10.7717/peerj.18238#supplemental-information.

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
