# Peer review of "Development of multiplex real-time PCR for simultaneous detection of common fungal pathogens in invasive mycoses"

_PeerJ, doi:10.7717/peerj.18238_

## Round 0.1 · original submission · Major Revisions

Please address issues pointed by both reviewers and revise manuscript accordingly

Reviewer 1 ·

Basic reporting

Review report of the article titled “Development of Multiplex Real-Time PCR for Simultaneous Detection of Common Fungal Pathogens in Invasive Mycoses”
The research work is really promising and can be used to be a good diagnostic for the diagnosis of fungal infections but I have some questions as below:

1. The multiplex PCR protocol given in this study was only evaluated in fungal pure cultures and not in clinical samples which is the major drawback of this study. Unless and until this results is not validated or reproduced in clinical samples, this results has less utility. In multiple places, authors stated that this test saves time of diagnosis which can only be said when tested directly in clinical samples. This work can be revised with patient samples validation.
Such diagnostic tests must be evaluated in clinical specimens to check cross reactivity and efficiency because lots of real challenges such as very low copy number, numerous PCR inhibitors etc. are dealt for clinical samples which is not present in case of pure culture.

2. The number of target and non-target fungal strains (clinical and ref.) evaluated are too small in numbers (example. Only 3 A. fumigatus, 3 A. terreus, 1 A. flavus etc) to come to the conclusion of an efficient PCR assay which can be further applied for clinical use.

3. Why A. flavus, the second most common species of invasive aspergillosis worldwide, and others like C. parapsilosis (prevalent worldwide) or C. tropicalis (prevalent in tropical climate) were not evaluated? These pathogens are of great importance because of rapidly emerging antifungal resistance among these species.

4. Line 50-51: the conventional sequencing methods takes 3-4 days to get the report, definitely not two weeks.

5. Line 210: No new relevant modification was done in this study as claimed by the authors. Bead beating is not necessary for molds DNA extraction. Multiple published articles show it can be achieved using simple high-speed vortexing or in case of using liquid nitrogen freezing method followed by lysis buffer and motor-pestle crushing, no vortexing is not even required.
Hence the DNA extraction method can be shortened proving suitable references.

6. Line 236: Not 28 targeted species- it should be 28 targeted strains.

Experimental design

Experiment designs are good until standerdization but more validation are needed with patient samples

Validity of the findings

The preliminary findings are good and promising but unless and until this results is not validated or reproduced in clinical samples, this results has less utility. In multiple places, authors stated that this test saves time of diagnosis which can only be said when tested directly in clinical samples. This work can be revised with patient samples validation.Unless and until this results is not validated or reproduced in clinical samples, this results has less utility. In multiple places, authors stated that this test saves time of diagnosis which can only be said when tested directly in clinical samples. This work can be revised with patient samples validation.

Additional comments

It would be a great work if the authors will validate this finding with clinical samples

·

Basic reporting

This study aims to provide a rapid, sensitive, and specific detection method of real-time PCR assay for the diagnosis of Aspergillus and Candida.

However, the manuscript presents several errors that requere major revisions by the authors, including:

In the Introduction Section:
- Include descriptions and uses of other diagnostic tools to highlight the importance of timely pathogen identification.

- Incorporate more epidemiological evidence about the significance of Candida and Aspergillus species selected in the study, more specifically A. terreus and C. glabrata compared to other species.

In the References sections:
- Use references that align with the information presented. For example:
-- Line 33 to 36): Not related to the topic.
-- Line 198 to 199: It would be better to use a review about false negatives and the importance of using an internal amplification control (IAC) in PCR reactions.
-- Line 223 to 225: The reference is not the most appropriate for this context.

- Paragraph (lines 41 to 42): Does not appear in the "References" section.

- Use reviews on the advantages of NAATs tests and their use in conventional diagnostics.

General recommendations:
- Improve english styles. Some paragraphs are grammatically confusing and need revision for clarity

- In the supplemental table S1. Use the full name of all microorganisms. In the case of fungi, update with the current taxonomic guide recommended. (Fungal Nomenclature: Managing Change is the Name of the Game. DOI: 10.1093/ofid/ofac559)

- Use abbreviations appropiately throughout the manuscript.

Experimental design

In the Methodology section:
According to the description, no standardization and optimization processes for the concentrations of primers and probes were performed or is not presented in the manuscript, which is recommended to increase yield and avoid reagent excess and the formation of secondary structures in the reaction. This also applies to the thermal profile.

In the Results section:
- It is possible to use different colors for each signal fluorophore (FAM, HEX, TEXRD, CY5.5) to facilitate the visibility of results.

- Correct the copies/µl values, as they should be in the scale of 10^5 (1ml is equivalent to 10^-3 µl).

- The high variability obtained at low concentrations with A. terreus and C. albicans could pose a risk of false-negative results with biological samples, considering that the Ct values were 38.54 and 34.90, respectively. It is recommended to address this issue in the document, from the perspective of fungal load in patients.

Validity of the findings

Although the results obtained with the standard curve are adequate, with slope values ​​between -3.1 to -3.6, as recommended in the guidelines. There are several points not clear about the standarization and optimization process that are required to ensure the actual performance of the reaction.

In the Discussion Section:
- It is not clear why the internal control gene glmM from H. pylori was used, considering the lack of relation between this pathogen and pulmonary infections. Likewise, the authors did not mention in the methodology section if all samples were spiked with control DNA to ensure amplification and the use of an IAC.

- Line 216 to 219: On the contrary, a slope within the range of -3.6 to -3.1 indicates a standard and accurate reaction, due to prior optimization of reaction conditions. This optimization process was not mentioned by the authors in the manuscript.

- Line 223 to 225: These assays were not shown in the results or methodology sections.

- Line 230 to 232: From a biological point of view, what is the real importance of detecting in a range of thousand copies/ml? Include references that support this perspective. Will it be necessary to improve the reaction conditions and optimization process?

Additional comments

N.A.

---

## Round 0.2 · accepted · Accept

All issues pointed out by the reviewers were adequately addressed and the manuscript was amended accordingly. Therefore, the revised manuscript is acceptable now.